# Common Bean (*Phaseolus vulgaris* L.) NAC Transcriptional Factor PvNAC52 Enhances Transgenic *Arabidopsis* Resistance to Salt, Alkali, Osmotic, and ABA Stress by Upregulating Stress-Responsive Genes

**DOI:** 10.3390/ijms25115818

**Published:** 2024-05-27

**Authors:** Song Yu, Mingxu Wu, Xiaoqin Wang, Mukai Li, Xinhan Gao, Xiangru Xu, Yutao Zhang, Xinran Liu, Lihe Yu, Yifei Zhang

**Affiliations:** 1College of Agriculture, Heilongjiang Bayi Agricultural University, Daqing 163319, China; yusong@byau.edu.cn (S.Y.); byndwmx@163.com (M.W.); byndwxq@163.com (X.W.); 15545653789@163.com (M.L.); gxh15652003@163.com (X.G.); 18822901166@163.com (X.X.); 13904567536@163.com (Y.Z.); lxr12132002@163.com (X.L.); 2Key Laboratory of Low-Carbon Green Agriculture in Northeastern China, Ministry of Agriculture and Rural Affairs, Daqing 163319, China; 3Heilongjiang Provincial Key Laboratory of Modern Agricultural Cultivation and Crop Germplasm Improvement, Daqing 163319, China

**Keywords:** common bean, NAC transcription factors, saline-alkali, osmotic, ABA, abiotic stress, reactive oxygen species, proline, cis-acting element

## Abstract

The NAC family of transcription factors includes no apical meristem (NAM), *Arabidopsis thaliana* transcription activator 1/2 (ATAF1/2), and cup-shaped cotyledon (CUC2) proteins, which are unique to plants, contributing significantly to their adaptation to environmental challenges. In the present study, we observed that the PvNAC52 protein is predominantly expressed in the cell membrane, cytoplasm, and nucleus. Overexpression of *PvNAC52* in *Arabidopsis* strengthened plant resilience to salt, alkali, osmotic, and ABA stresses. PvNAC52 significantly (*p* < 0.05) reduced the degree of oxidative damage to cell membranes, proline content, and plant water loss by increasing the expression of *MSD1*, *FSD1*, *CSD1*, *POD*, *PRX69*, *CAT*, and *P5CS2*. Moreover, the expression of genes associated with abiotic stress responses, such as *SOS1*, *P5S1*, *RD29A*, *NCED3*, *ABIs*, *LEAs*, and *DREBs*, was enhanced by *PvNAC52* overexpression. A yeast one-hybrid assay showed that PvNAC52 specifically binds to the cis-acting elements ABRE (abscisic acid-responsive elements, ACGTG) within the promoter. This further suggests that PvNAC52 is responsible for the transcriptional modulation of abiotic stress response genes by identifying the core sequence, ACGTG. These findings provide a theoretical foundation for the further analysis of the targeted cis-acting elements and genes downstream of PvNAC52 in the common bean.

## 1. Introduction

Throughout their development and growth, plants encounter diverse abiotic stresses, including low temperature, drought, high temperature, salinity, and heavy metals, which may affect their productivity, development, and growth [1]. Plants have developed various response mechanisms and strategies to combat the negative effects of abiotic stress [2]. Transcription factors (TFs) play essential roles in plants’ adverse responses. When plants are subjected to unfavorable environmental conditions, they activate TFs through a series of signal transduction pathways, and TFs then modulate the expression of stress-responsive genes via interactions with cis-elements [3]. The NAC family of TFs includes no apical meristem (NAM), *Arabidopsis* transcription activator 1/2 (ATAF1/2), and cup-shaped cotyledon (CUC2) proteins, and the NAC TFs are highly conserved at the N-terminus and contain similar DNA-binding structural domains; however, the C-terminus is highly varied and lacks any recognized structural domains [4,5,6].

According to previous studies, NAC TFs play a role in plant development and growth, as well as in mediating the modulation of phytohormones and biotic and abiotic stress response processes [7,8,9]. When plants are subjected to abiotic stresses, NAC acts as both an activator and a repressor of TFs, resulting in tolerant or sensitive phenotypes in plants by up or downregulating the expression of downstream target genes. For example, in *Arabidopsis thaliana* (hereafter called *Arabidopsis*), ANAC019, ANAC055, and ANAC072 are positive regulators of abiotic stresses, including high salt levels, drought, ABA, and others [10]; *Arabidopsis* TFs *NAC016* and *AtNAP* function as negative modulators in drought and salt stress by repressing *AREB1* transcription [11,12]. In soybeans, oGmNAC085 overexpression enhances drought tolerance [13], whereas GmNAC11 overexpression increases salt sensitivity in transgenic plants [14]. OsNAC2 regulates salt tolerance and rice drought [15]. The overexpression of *MbNAC25* in *Bauhinia* spp. improves salt and cold tolerance in *Arabidopsis* [16]. Additionally, *ZmNAC071*, a transcriptional repressor, improves the sensitivity of transgenic *Arabidopsis* plants to osmotic and ABA stress by downregulating stress-responsive genes [17]. However, research focusing on the functional characterization of NAC TFs in the common bean (*Phaseolus vulgaris* L.) is scarce.

Stress—e.g., temperature, salinity, drought, or metals—triggers the excessive and rapid accumulation of reactive oxygen species (ROS) in plants, resulting in oxidative stress [18]. Plants have evolved complex regulatory systems to efficiently regulate intracellular ROS redox homeostasis [19]. For example, NAC TFs play vital roles in the regulation of ROS metabolism in the presence of abiotic stress. For example, *SbNAC9* enhances drought tolerance in sorghum by scavenging ROS and activating genes associated with the stress response [20], and rice *OsNAC3* enhances heat and drought tolerance by regulating the ROS dynamic balance [21]. *ZmNAC074* overexpression in transgenic *Arabidopsis* could regulate the accumulation of various stress metabolites, such as ROS, antioxidants, and malondialdehyde (MDA), by inducing the expression of ROS-scavenging genes to enhance the heat tolerance of maize plants under high-temperature stress conditions [22]. In response to osmotic and salt stress, BpNAC012 enhances heat tolerance by upregulating the expression of *SODs* and *PODs* genes in *Betula* [23]. In recent times, as people’s living standards and consumption patterns have gradually improved, the global demand for the common bean has been increasing every year; however, various abiotic factors often affect the yield and quality of the common bean [24,25,26], Therefore, continuous in-depth and systematic investigation of the common bean’s responses to abiotic stress mechanisms and tolerance enhancement regulatory pathways is of great significance. However, in abiotic stress, it remains uncertain whether NACs also play an essential role in ROS metabolism.

The present study aimed to investigate how common bean NAC TFs regulate plant responses to various abiotic stresses, such as saline-alkali, ABA, and osmotic stress. To this end, a novel transcriptional activator, *PvNAC52*, was screened from the common bean ‘Qingyun No. 1’, and its functional characterization was investigated. The results indicated that the overexpression of *PvNAC52* in *Arabidopsis* increased *SOD*, *MSD1*, *FSD1*, *CSD1*, *POD*, *PRX69*, *CAT*, *P5CS2*, *SOS1*, *P5S1*, *RD29A*, *NCED3*, *ABI*, *DREB*, and *LEA* expression, improving ROS scavenging, proline accumulation, and abiotic stress-related genes, which in turn enhanced the plant’s resistance to salt-alkali, ABA, and osmotic stress. The cis-acting element ABRE, ACGTG, was found to play a significant role in the regulation of PvNAC52-mediated gene expression. Based on these results, we enhanced the understanding of how common bean NAC TFs modulate plant responses to diverse abiotic stresses, including salt-alkali, ABA, and osmotic stresses.

## 2. Results

### 2.1. PvNAC52 Gene Cloning and Sequence Analysis

Through an analysis of transcriptome data (SRX2672484), a presumed abiotic-stress-related NAC gene Phvul.005G084500 was isolated from a common bean of the ‘Qingyun No. 1’ variety (ID: 18630470); the gene sequence is shown in Appendix A. We named it *PvNAC52* according to its position on the chromosome. The open reading frame of *PvNAC52* was 1023 bp and encoded 340 amino acids.

According to protein sequence alignments, PvNAC52 has two domains: an NAC domain at the N-terminus, which is relatively conserved among dicotyledons, and a variable domain at the C-terminus (Appendix A). The NAC domain has five subdomains. The N-terminal domain consists of a short chain of basic amino acids (PRDRKYP) that may function as a nuclear localization signal (NLS). The relationship between PvNAC52 and NAC was investigated by constructing a phylogenetic tree. The results showed that PvNAC52 had high homology with NACs in dicotyledonous plants but low homology with NACs in monocotyledonous plants (Appendix A).

### 2.2. Expression and Protein Localization of PvNAC52 Gene

Using the leaves of common bean seedlings treated with different types of salt and alkali stress, qRT-PCR analyses were performed to investigate *PvNAC52* expression characteristics. *PvNAC52* expression was markedly upregulated (*p* < 0.01) in response to neutral and basic salt stress (Figure 1A,B). Under alkaline salt stress, *PvNAC52* expression showed a sustained continuous increase and peaked after treatment for 48 h (Figure 1A). Following neutral salt stress treatment, *PvNAC52* expression was first upregulated and then downregulated, reaching a peak value after 24 h of treatment (Figure 1B). According to these results, *PvNAC52* may play an essential role in the common bean’s responses to abiotic stresses, including those caused by salt and alkali presence.

To assess the subcellular localization of PvNAC52, pBI121-PvNAC52-EGFP and pBI121-EGFP vectors were transferred into wild-type (WT) *Arabidopsis*, and a transgenic plant was generated. The roots of a stable transformant were visualized using a confocal laser microscope, and the roots of *Arabidopsis* containing pBI121-PvNAC52-EGFP fusion proteins showed the expression in the cell membrane, cytoplasm, and nucleus (Figure 1C).

### 2.3. Analysis of PvNAC52 Transcriptional Activation

To determine whether PvNAC52 demonstrated transcriptional enhancement capabilities, the full-length coding sequence (CDS) of PvNAC52 was separated into the PVNac52-N-terminal (amino acids 1–160) and PVNAC52-C-terminal (amino acids 161–340) sequences for yeast transcriptional activity determination. It was found that all transformers grew normally in SD/-Trp and SD/-Trp media containing X-α-Gal. However, on the SD/-Trp medium supplemented with X-α-Gal, the full-length CDS PVNAC52-C-terminal NAC domain, and the positive control pGBKT7-53 turned blue, indicating that the PvNAC protein showed transcriptional self-activation activity localized in the C-terminal region. Further verification in SD/-Ade/-Trp/-His/X-alpha-Gal medium showed that only the yeast containing full-length CDS, the NAC domain at the PvNAC52-C terminal, and the positive control pGBKT7-53 grew normally and turned blue. However, yeast cells containing the NAC domain on the pGBKT7 and PvNAC52-N terminals did not grow normally (Figure 2). It appears that transcriptional self-activation activity is exhibited by the PvNAC52 protein, with the activation site localized specifically in its C-terminal region.

### 2.4. PvNAC52 Positively Regulates Plant Responses to Saline-Alkali, ABA, and Osmotic Stress

Eight independent T3 homozygous lines with *PvNAC52* overexpression were detected and evaluated by qRT-PCR. In accordance with *PvNAC52* expression (Appendix A), overexpression transgenic lines three and five (OE3/OE5) with high *PvNAC52* expression level were chosen, along with an empty control vector line (35S) and WT control line plants. Under normal culture conditions, the four lines did not differ significantly according to fresh weight, root length, or germination rate. Under neutral salt, alkaline salt, ABA, and mannitol stress, OE plants showed notably increased germination rates, fresh weights, and root lengths compared to those of the WT and 35S plants (*p* < 0.01, *p* < 0.05; Figure 3A–E).

Simultaneously, the resistance of plants with *PvNAC52* overexpression to neutral salt, basic salt, ABA, and mannitol at the seedling stage under soil culture conditions was determined. Under normal conditions, OE plants grew similarly to the WT and 35 lines (Figure 4A–E); however, under four different stress treatments, OE plants grew better than the WT and 35S plants, had higher chlorophyll content, and had lower water loss rates than the control lines (Figure 4F,G), and both findings were highly significant (*p* < 0.01). In addition, root analysis at the seedling stage revealed that under normal conditions, the root growth state of the OE plant was relatively similar to that of the WT and 35S plants. Nonetheless, after 14 days of NaCl, NaHCO_3_, ABA, and mannitol stress treatments, the roots of OE plants were notably (*p* < 0.01) longer and had more root tips than the WT and 35S plants (Appendix A). Under NaCl, NaHCO_3_, and mannitol treatments, the root surface areas of OE plants were significantly greater (*p* < 0.01) compared to the WT and 35S plants (Appendix A). In conclusion, PvNAC52 can improve plant resistance to saline-alkali, ABA, and mannitol stress at different growth stages.

### 2.5. Effect of PvNAC52 on Lipid Peroxidation Level of Cell Membrane

Based on the propidium iodide (PI) and Evans blue staining results, as well as the changes in MDA content and conductivity, this study analyzed whether *PvNAC52* could affect the lipid peroxidation level of cell membranes under different stress treatments. No obvious differences were noted among the OE, WT, and 35S plants under normal conditions (Figure 5A,B), but under neutral salt, basic salt, ABA, and mannitol stress, the OE plants stained considerably less than the WT and 35S plants. In addition, under different stress treatments, the MDA content and conductivity of OE plants were markedly (*p* < 0.01) declined compared to those of the WT and 35S plants (Figure 5C,D). Based on these results, *PvNAC52* alleviated the severity of cell membrane damage caused by salt-alkali, ABA, and mannitol stresses.

### 2.6. Regulatory Effect of PvNAC52 on Reactive Oxygen Species Clearance System

First, NBT and DAB staining methods were used to investigate *PvNAC52*’s effect on the ROS balance in plants under salt-alkali, ABA, and mannitol stress. The accumulation levels of hydrogen peroxide (H_2_O_2_) and superoxide anion radical (O_2_^·−^) of *PvNAC52*-overexpressed lines and the WT and 35S plants under various stress conditions were compared. Under normal culture conditions, no obvious differences were observed among the WT, 35S, and OE plants in DAB and NBT staining. However, after NaCl, NaHCO_3_, ABA, and mannitol treatments, DAB and NBT staining of the OE plant was shallower than that of the WT and 35S plants (Figure 6A,B). The root tip test results of seedlings stained with H_2_DCF-DA also showed that under the four stress treatments, the root tip color of the OE plant was remarkably lower than that of the WT and 35S plants (Figure 6C). We also found that the H_2_O_2_ content of *PvNAC52*-overexpressed lines after NaCl, NaHCO_3_, ABA, and mannitol treatment was noticeably lower (*p* < 0.01) compared of the WT and 35S plants (Figure 6D), in agreement with the staining results.

Furthermore, we compared the activities of catalase (CAT), peroxidase (POD), and superoxide dismutase (SOD) in overexpressed *PvNAC52* lines and in WT and 35S plants under different stress conditions. The OE plant exhibited significantly (*p* < 0.01) higher CAT, POD, and SOD activities than the WT plant under NaCl, NaHCO_3_, ABA, and mannitol treatments (Figure 6E–G). Given the positive effects of *PvNAC52* on POD, SOD, and CAT activities in plants exposed to different stress treatments, we assessed the effects of *PvNAC52* on *SOD, POD*, and *CAT* gene expression. Under normal conditions, the expression levels of *SOD* (*SOD*, *CSD1* and *FSD1*), *POD* (*POD*, *PRX69*), and *CAT* genes in the *PvNAC52*-overexpressed line were comparable with those in the WT and 35S plants. Nevertheless, under various stress treatments, *SOD*, *POD,* and *CAT* expression was notably higher (*p* < 0.01) in the OE plant than that in the WT and 35S plants (Figure 7). Further evidence suggests that *PvNAC52* positively regulates the ROS balance mediated by *SOD*, *POD,* and CAT under abiotic stress conditions.

### 2.7. Analyzing the Regulation of Proline Metabolism by PvNAC52

The proline contents of the WT and PvNAC52 overexpression lines were relatively similar under normal culture conditions. However, after treatment with NaCl, NaHCO_3_, ABA, and mannitol, the proline content of all lines remarkably increased compared to that under normal culture conditions, and the OE plants had a notably (*p* < 0.01) greater proline content than the WT and 35S plants (Figure 8A). The expression of one proline biosynthesis gene (*P5CS2*) and two proline degradation genes (*ProDH* and *P5CDH1*) were detected. As shown in Figure 8B–D, normal culture conditions did not significantly differ in the expression levels of OE, WT, and 35S plants; however, after NaCl, NaHCO_3_, ABA, and mannitol treatments, the OE plants expressed P5CS2 significantly higher (*p* < 0.01) than the WT and 35S plants. In contrast, *M5CDH1* and *ProDH* expression was markedly (*p* < 0.01) reduced in the OE plants under the four stress conditions.

### 2.8. PvNAC52 Promotes the Expression of Different Stress-Responsive Genes

The expression changes in 11 abiotic stress response-associated genes were determined using qRT-PCR to explore the regulatory pathway of the PVNAC52-mediated abiotic stress response. The results showed that the OE, WT, and 35S plants exhibited relatively similar expression levels of the 11 genes under normal culture conditions. Nonetheless, under stress treatment, the salt-stress-related genes *SOS1* and *P5CS1* (Figure 9A,B), the alkali-stress-affected genes *RD29A* and *NCED3* (Figure 9C,D), the ABA-pathway-related genes *ABI3* and *ABI5* (Figure 9E,F), and the drought-stress-affected genes *AtDREB1A* and *AtDREB2A* (Figure 9G,H), as well as three adversity-responsive genes, *AtLEA3*, *AtLEA7*, and *AtLEA14* (Figure 9I–K), were noticeably higher in expression in the OE plants than in the WT and 35S plants (*p* < 0.05, *p* < 0.01).

### 2.9. Combination Evaluation of PvNAC52 Cis-Acting Elements

*PvNAC52* (Appendix A) promoter cis-acting elements were estimated using the PlantCARE database, and 28 cis-acting elements associated with abiotic stress were observed within the 1500 bp promoter region upstream of the transcription start site of *PvNAC52*. There were 10 abscisic acid responsive elements (ABREs), five G-boxes (light responsive element), four MYBs, four MYCs, three jasmonic acid responsive elements (CGTCA-motif), and one each of TC-rich and anaerobic response element (ARE) components. The yeast hybridization (Y1H) system was used to further validate the sequence binding of PvNAC52 to the most abundant ABRE element in its promoter. First, the minimum Aureobasidin A (AbA) resistance of each bait-carrier strain was measured. The results showed that all strains grew normally when AbA concentrations were 0, 100, and 150 ng, but p53 AbAi could not grow normally when the AbA concentration was greater than or equal to 200 ng, and P53 abai could not grow normally when the ABA concentration was greater than or equal to 250 ng. pAbAi-ABRE did not grow normally, whereas pAbAi-MABRE1 and pAbAi-MABRE2 failed to grow normally at 300 ng (Figure 10A). Therefore, the maximum AbA concentration at which different bait strains can grow normally was selected as the AbA addition concentration of the SD/-Leu plate, and the prey vector containing PvNAC52 was transformed into the yeast reporter strain containing the ABRE element sequence and its mutant sequence bait vector. To observe the growth status of the strains, the co-transformation of pGADT7-p53 and p53-AbAi was used as a positive control. All transformed strains grew normally in SD/-Leu medium without AbA (Figure 10B), whereas yeast cells containing only the ABRE elements and the positive control, p53-AbAi, grew normally in plates containing the corresponding AbA concentrations. Yeast cells containing the mutated elements MABRE1 and MABRE2 exhibited atypical growth patterns. These findings demonstrate that PvNAC52 binds to the cis-acting element ABRE in yeast but cannot bind to its mutant sequence.

## 3. Discussion

The activation mechanism of the NAC-protein-mediated abiotic stress response was further investigated by identifying various NAC-type transcriptional activators [21,27,28,29,30]. Studies have demonstrated that some NAC-type TFs regulate abiotic stress responses [11,12]. In this study, we identified PvNAC52, an NAC transcriptional activator, in the common bean. *PvNAC52* contains an ORF of 1023 bp that encodes 340 amino acids. Protein sequence alignment revealed that the N-terminal region of the *PvNAC52* protein not only harbored a conserved NAC domain but also featured the LVFY motif (Appendix A). Prior research has demonstrated that certain NAC proteins possess a conserved NAC repression NARD-like domain within their N-terminal regions, and that the LVFY motif within this NARD-like domain is deemed crucial for exerting transcriptional inhibition activity. Finally, the functions of NAC TFs containing both transcriptional regulatory and inhibitory domains rely on interactions between these two domains [14,17]. However, in this study, the full-length and C-terminal *PvNAC52* fragment grew normally on the selective medium, indicating the activation of TFs in the C-terminal region (Figure 2). This suggests that *PvNAC52* acts as a TF, possibly because the C-terminal transcriptional activation structural domain is more powerful than the NARD-like structural domain within the N-terminal region.

Based on phylogenetic analysis, *PvNAC52* was highly related to soybeans, cowpeas, and other legumes (Appendix A). Subcellular localization analysis of pBI121-PvNAC52-GFP revealed that *PvNAC52* was mainly localized in the cell membrane, cytoplasm, and nucleus (Figure 1C). Although the roles of NAC TF in modulating antioxidant systems under abiotic stress have been assessed in diverse species, the effects of NAC TFs on the common bean’s response to stress have rarely been reported. In this study, we compared the phenotypic differences between *Arabidopsis* plants overexpressing *PvNAC52* and WT-type *Arabidopsis* lines under salt, alkaline, ABA, and mannitol stress and observed that the germination rate, seedling root length, whole-plant fresh weight, and chlorophyll content of rosette leaves of *Arabidopsis* plants overexpressing PvNAC52 were significantly improved (Figure 3A–E and Figure 4A–F). *AtMYB61* overexpression has been shown to increase drought tolerance in *Arabidopsis* by reducing stomatal pore size [31], whereas some abiotic-stress-related transcription factors, such as *BplMYB46* [32], *AST1* [33], and *ZmNAC071* [17], can positively or negatively regulate *AtMYB61* or its homologous genes, thereby reducing or increasing the stomatal pore size in an abiotic stress environment. However, under normal culture conditions, the transpiration rate of overexpressed *PvNAC52* lines decreased (Figure 4G). Whether this was due to the upregulation of *AtMYB61* expression by *PvNAC52* and the reduction in stomatal pore size, which led to a reduction in water loss in *Arabidopsis*, remains to be further investigated.

Under normal growth conditions, ROS production and clearance in plants maintain a dynamic balance; however, under stress conditions, ROS levels increase, resulting in the oxidation of lipids, proteins, and nucleic acids in cell membranes [34,35]. In this study, when stressed with salt, alkali, ABA, and mannitol, overexpressed *PvNAC52* lines produced lower ROS accumulation levels than WT plants; however, they produced higher levels of CAT, POD, and SOD (Figure 6A–G). Expression analysis of antioxidase-associated genes revealed that they showed significantly higher expression in the *PvNAC52* line compared to the WT plants after each stress treatment (Figure 7). This suggests that *PvNAC52* inhibits excessive ROS accumulation by regulating genes downstream of the ROS metabolic pathway, which in turn increases CAT, POD, and SOD activities. Additionally, compared with WT, plants overexpressing *PvNAC52* showed less severe cell membrane damage and lower levels of MDA under different stress conditions (Figure 5A–D). The findings of our study suggest that *PvNAC52* derived from the common bean can actively modulate the antioxidant system of plants under extreme conditions, thereby exerting a positive regulatory effect.

Proline is the most common endogenous osmotic substance that accumulates under various abiotic stresses, effectively removes reactive oxygen species, and is a vital indicator of abiotic stress resistance in plants [36,37,38]. *P5CDH1* and *ProDH* are related to proline degradation in *Arabidopsis*, and *P5CS2* is a proline biosynthesis gene [39,40]. Therefore, the proline content and expression of *P5CDH1*, *ProDH,* and *P5CS2* in plants under salt, alkali, ABA, and mannitol stress were analyzed in this study. Under normal conditions, no obvious differences in proline content were observed among the OE, WT, and 35S plants. However, after stress treatment, the OE plant exhibited higher proline content than the WT and 35S plants (Figure 8A). Additionally, after different abiotic stress treatments, *ProDH* and *P5CDH1* expression levels in the OE plants were remarkably decreased compared to the WT and 35S plants, whereas *P5CS2* expression was obviously decreased in the OE plants compared to the WT and 35S plants (Figure 8B–D). These findings indicate that the overexpression of *PvNAC52* increased proline content in transgenic lines by inducing the upregulation of proline biosynthesis genes and downregulation of proline-degradation-related genes, thereby enhancing the saline, alkali, ABA, and mannitol tolerance in transgenic plants.

Research has shown that the transcription factor *FcWRKY40* in *Fortunella crassifolia* can directly regulate *FcABF2* and *FcP5CS1* and increase proline biosynthesis, thereby improving plant salt tolerance [41]. Soybean *GmMAX2a* improves drought resistance and saline-alkali tolerance in *Arabidopsis* by regulating stress-associated genes, including *RD29A* and *NCED3* [42]. *Arabidopsis LSH8* positively regulates the expression levels of ABA-signaling-associated genes *ABI5*, *ABI3*, *RAB18*, and *RD29B* in response to ABA stress [43]. *SiCDPK24* improves *Arabidopsis* survival under drought stress by modulating genes related to drought stress, such as *CBF3*/*DREB1A* and *DREB2A* [44]. LEA proteins have been widely implicated in seed development and adaptation to diverse abiotic stresses in higher plants [45]. Therefore, eight genes (*SOS1*, *P5CS1*, *RD29A*, *NCED3*, *ABI3*, *ABI5*, *AtDREB1A*, *AtDREB2A*) related to the response to salt, alkali, ABA, and mannitol stress were selected. Three *LEA* genes (*AtLEA3*, *AtLEA7*, *AtLEA14*) with stress-resistance functions further verified the regulatory roles of *PvNAC52* in plant responses to abiotic stresses. The results showed that, following different stress treatments, the OE plants expressed significantly higher levels of the above stress-associated genes than the WT or 35S plants (Figure 9). In conclusion, the upregulation of these stress response genes in *PvNAC52* transgenic lines enhanced plant tolerance to abiotic stresses, further supporting the role of *PvNAC52* as a positive regulatory promoter.

TFs can function as major regulators of genes associated with various environmental stressors, and each binds to a distinct cis-regulatory element within the promoter, creating a distinct stress response network [46]. Many TFs can bind to elements that modulate the function of downstream genes; for example, DREB1/CBF transcription factors control the expression of downstream stress response genes and regulate plant tolerance to osmotic stress by binding to DRE/CRT cis-regulatory elements within gene promoters [47]. *OsABI5* binds to G-box elements and activates reporter gene expression [48]. It has been shown that ABA serves as a modulator to induce various responses within certain stress signaling pathways [49], and that *Arabidopsis* [10,50,51], rice [52], soybean [14], cucumber [29], wheat [53,54] and maize [17] transcription factors, such as bZIP, NAC, etc., can be involved in modulating the stress response through ABA-dependent/-independent signaling pathways. In this study, we investigated the promoter sequence of *PvNAC52* and found that ABRE (an ABA-responsive element) is the most cis-acting element within the promoter. Therefore, yeast single hybridization was used to explore whether *PvNAC52* combined with the ABRE element, and the findings demonstrated that the *PvNAC52* protein could interact with ABRE but not with ABRE mutant elements (Figure 10). We further confirmed that *PvNAC52* bound to ABRE cis-acting elements, participated in the ABA regulation pathway, and activated downstream resistance genes.

## 4. Materials and Methods

### 4.1. Plant Components and Stress Treatment

The common bean variety used in this study was ‘Qingyun No. 1’, a salinity-tolerant common bean variety selected by Heilongjiang Bayi Agricultural and Reclamation University. Firstly, seeds with uniform and full grains were chosen, and the surface of the seeds was sterilized with 10% sodium hypochlorite solution and then cultivated at 25 for 3 d to germinate. Next, seeds with neat and uniform germination were selected and transplanted into a tank containing 1/4 Hoagland nutrient solution for cultivation, and the nutrient solution underwent aeration from an air pump via an air stone on an hourly basis. The hydroponic conditions included a temperature of 25 °C, a light intensity of 400 μmol·m^−2^·s^−1^, and photoperiods of 14 h light incubation and 10 h dark incubation. After 5 d of incubation, the plants were subsequently placed in 1/2 Hoagland nutrient solution to continue incubation; when the brassicas had grown to the point where the second pair of compound leaves were fully expanded, they were transplanted to the Hoagland total nutrient solution, and 100 mmol·L^−1^ NaCl solution (neutral salt treatment) and 100 mmol·L^−1^ NaHCO_3_ (alkaline salt treatment) were mixed with the culture solution. This was followed by incubation for 0, 24, and 48 h (with 0 h treatment as the control). At 0, 24, and 48 h (with treatment 0 h as the control), the second fully expanded compound leaves of the common bean seedlings were harvested, snap-frozen, and preserved at −80 °C.

The ecotype of *Arabidopsis* used as a WT was Colombia-0, and the seeds were cultivated on MS medium for 7 days. After culturing *Arabidopsis* seeds on MS medium for 1 week, the seedlings were transplanted into a nutrient bowl containing black soil and vermiculite (volume ratio of 1:1) and placed in a greenhouse with a day and night temperature of 22 °C/16 °C for cultivation. When the plants were 4 weeks old, the WT, OE, and 35S *Arabidopsis* plant lines were exposed to deionized water (control), 10 mM NaHCO_3_, 180 mM NaCl, 50 μM ABA, and 300 mM mannitol, respectively. After 48 h of treatment, *Arabidopsis* rosette leaves were harvested from the rosettes under each treatment condition.

### 4.2. qRT-PCR Detection

Total RNA was extracted using TRIzol^®^ Reagent (Invitrogen, Carlsbad, CA, USA) and assayed for RNA quality using a NanoDropTM One Microvolume UV-Vis spectrophotometer (Thermo Fisher Scientific, Waltham, MA, USA) and 1% gel. Then, cDNA synthesis was conducted using a ReverTra Ace^®^ qPCR RT Master Mix with gDNA Removal kit (TOYOBO Co., Ltd., Osaka Prefecture, Japan). Primers were designed using Primer Premier software (version 5.0) (https://www.statistical-analysis.top/Primer/Details.html, accessed on 10 January 2022), with *PvACT11* (GenBank: CV529679) and *PvIDE* (GenBank: FE702602) employed as internal genes to determine *PvNAC52* expression [55], and *AtACT11* (Gene ID: AT5G09810) and *AtTUB2* (Gene ID: AT5G62690) used as reference genes to probe target gene expression in *Arabidopsis thaliana* [56]. Information on all primer pairs used for quantitative real-time PCR experiments is detailed in Appendix A. Real-time qRT-PCR reaction system and conditions were performed using the SYBR^®^ Green Real-time PCR Master Mix kit (TOYOBO Co., Ltd., Osaka Prefecture, Japan) instructions. The data were calculated using the 2^−ΔΔCt^ approach [57].

### 4.3. Assessment of Subcellular Localization

The full-length CDS of *PvNAC52* with the stop codon deleted was integrated into the pBI121-EGFP vector (see Appendix A for primer details). The recombinant vector (pBI121-PvNAC52-EGFP) and empty control vector (pBI121-EGFP) were stably transformed into *Arabidopsis* using the floral dip approach. The root systems of transgenic *Arabidopsis* seedlings cultivated on ½ MS medium for 1 week were examined, and the fluorescence intensities of the lower epidermis of the transgenic plants were observed using a TCS SP8 laser confocal microscope (Leica Microsystems Inc., Wetzlar, Germany).

### 4.4. Transcription Activation Test

The full-length N-terminal (amino acids 1–160) and C-terminal (amino acids 161–340) CDS sequences of *PvNAC52* (primers detailed in Appendix A) were used to insert PvNAC52, PvNAC52-N, and PvNAC52-C into the vector pGBKT7. Reference was made to the manufacturer’s manual for the Matchmaker Gold Yeast Two-Hybrid System (Clontech Laboratories, Inc., San Jose, CA, USA) to convert the different vectors into yeast lines. Finally, the yeast lines containing the transformation vector were cultured in SD/-Trp, SD/-Trp/X-α-Gal, SD/-Trp/X-α-Gal/AbA, and SD/-Trp/-Ade/-His/X-α-Cultivate on Gal solid medium for 3–5 d at 30 °C. By analyzing the growth of yeast lines containing transformation vectors in the culture medium, we determined whether they had transcriptional activation activity, using pGBKT7 and pGBKT7-53 as negative and positive controls, respectively.

### 4.5. Construction of Transgenic Arabidopsis Lines

PCR was used to amplify the entire coding sequence of *PvNAC52*, using the primers listed in Appendix A. The amplified sequence was introduced into the pBI121-EGFP vector. The expression of *PvNAC52* was regulated by the CaMV 35S promoter within the recombinant vector. Employing the floral dip technique, both the recombinant vector (35S PvNAC52) and empty pBI121 EGFP vector (35S) were transformed into *Arabidopsis*. Following transformation, homozygous lines were chosen on culture medium supplemented with kanamycin, a process that continued until the T3 generation was achieved.

### 4.6. Salt Alkali, Mannitol, and ABA Treatment

To investigate the effects of salt alkali, ABA, and mannitol on seed germination, the 35S, OE, and WT plants seeds were cultivated in 1/2 MS medium containing 120 mM NaCl, 7 mM NaHCO_3_, 1.5 μM ABA, and 300 mM mannitol, respectively, and the germination rates were determined statistically 8 d post-sowing. For the observation of plant seedling phenotypic differences, *Arabidopsis* seeds of 35S, OE, and WT plants were first cultivated in 1/2 MS medium, and following 7 d of incubation under normal conditions, the seedlings were transplanted into MS medium containing 120 mM NaCl, 7 mM NaHCO_3_, 1.5 μM ABA, and 300 mM mannitol and vertically incubated for 7 d. Finally, photographs were captured to visually assess and quantify both root length and fresh weight. To study the tolerance of plants to salt alkali, ABA, and mannitol during the seedling stage, soil-trained seedlings of 4-week-old WT, OE, and 35S plants were exposed to deionized water (control), 180 mM NaCl, 10 mM NaHCO_3_, 50 μM ABA, and 300 mM mannitol, and rosette leaves were collected at 5 d after treatment to determine the electrical conductivity. Photographs were taken 16 d after treatment, and the relevant physiological and biochemical indices of the leaves were determined.

### 4.7. Analysis of Relevant Physiological and Biochemical Indicators

In this study, the conductivity was determined with reference to the method previously published by Wang et al. [58]. SOD (nitrogen blue tetrazolium method), CAT (UV-absorbent method), and POD (guaiacol method) activities, as well as H_2_O_2_ (titanium sulfate colorimetric method), proline (acid ninhydrin method), and MDA (thiobarbituric acid method), were determined using kits derived from (Grace Biotechnology Co., Ltd., Suzhou, China). Chlorophyll was extracted using 70% acetone, and its content was determined based on a previously reported method [59].

### 4.8. Chemical Tissue Staining Analysis

WT, OE, and 35S plant seeds (4 weeks old) were taken and exposed to 180 mM NaCl, 10 mM NaHCO_3_, 50 μM ABA, and 300 mM mannitol for 2 h. The plants were then stained and analyzed. For propidium iodide (PI) staining, plants were infiltrated with 25 μg/mL PI solution for 30 min and finally photographed for observation using a laser confocal microscope (Leica TCS SP8). Evans blue staining was performed using the method previously published by Kim et al. [60]. H_2_DCF-DA, NBT, and DAB staining were performed in accordance with the method published by Zhang and co-workers [61].

### 4.9. Yeast One-Hybrid (Y1H) Assay

The 1500 bp promoter sequence upstream of the PvNAC52 transcription initiation site can be downloaded from the Phytozome database (https://phytozome-next.jgi.doe.gov/ (accessed on 24 January 2021)). PlantCARE (http://bioinformatics.psb.ugent.be/webtools/plantcare/html/ (accessed on 24 January 2021)) was utilized to estimate the adversity-associated cis-acting elements contained within the *PvNAC52* promoter, and then the element with the most occurrences in the promoter sequence was selected for the ABRE (ACGTG) for recognition element analysis. The *PvNAC52* CDS was inserted into the pGADT7 vector to obtain the effector vector (effector vector). Simultaneously, the NAC recognition and mutation element sequences (MABRE1:AAATG and MABRE2:AAAAG) were duplicated in tandem and integrated into the pAbAi vector. The primer pairs used are listed in Appendix A. Effector vectors were co-transformed with reporter vectors into Y1H yeast lines in accordance with the manufacturer’s instructions (Clontech Laboratories Inc., San Jose, CA, USA). The growth performance of co-transformed cells on solid culture media with or without Aureobasidin A (AbA) (SD/-Leu or SD/-Leu/AbA) was determined.

### 4.10. Statistical Analysis

Statistical analyses were conducted using SPSS v26.0 software. Three biological replicates were set independently for the analysis of plant growth and physiological and biochemical indicators. qRT-PCR was performed using three biological and technical replicates. All values are repeated averages (±SD). The experimental data were compared using Student’s *t*-test (* *p* < 0.05, ** *p* < 0.01).

## 5. Conclusions

In summary, the highly expressed *PvNAC52* gene was screened from saline-tolerant common beans (‘Qingyun No. 1’) under saline-alkali stress. This gene is mainly localized within the cell membrane, cytoplasm, and nucleus, has a transcriptional activation potential in the C-terminal region, and regulates downstream target genes by binding to the ABER cis-acting element. The overexpression of *PvNAC52* in *Arabidopsis* significantly enhanced its tolerance to saline-alkali, ABA, and mannitol stress. This enhanced resistance to abiotic stress was attributed to increased membrane stability, proline content, and ROS scavenging capacity, as well as reduced ROS levels, membrane damage, and water loss rate. Another possible cause is the altered expression of various genes related to the stress response and ABA signal transduction in *PvNAC52*-overexpressed plants (Figure 11). These strategies can work together to enhance the defense ability of transgenic lines against saline-alkali, ABA, and mannitol stress.

## Figures and Tables

**Figure 1 ijms-25-05818-f001:**
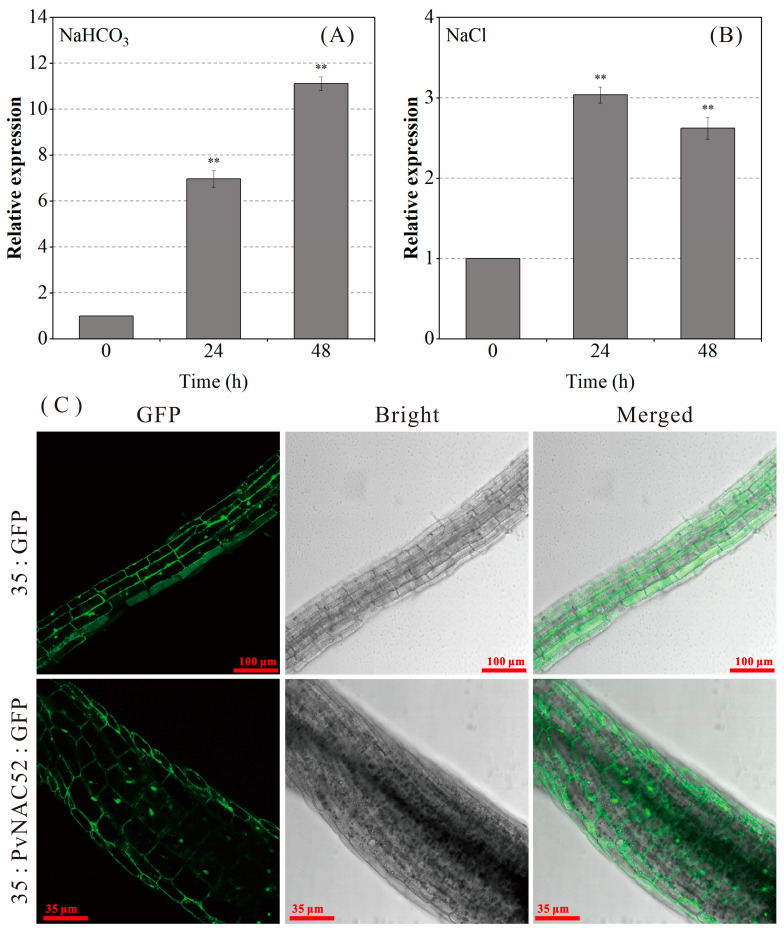
Expression and protein localization of PvNAC52. (**A**) *PvNAC52* gene expression profile of common beans under 100 mmol·L^−1^ NaHCO_3_ stress. (**B**) *PvNAC52* gene expression profile of common beans under 100 mmol·L^−1^ NaCl stress. (**C**) Localization of PvNAC52 in transgenic *Arabidopsis* root cells. *PvNAC52* expression normalized to one after 0 h of neutral or basic salt stress. Data are presented as the mean ± standard. Student’s *t*-test was used to compare the significant differences between each treatment time and 0 h treatment, ** *p* < 0.01.

**Figure 2 ijms-25-05818-f002:**
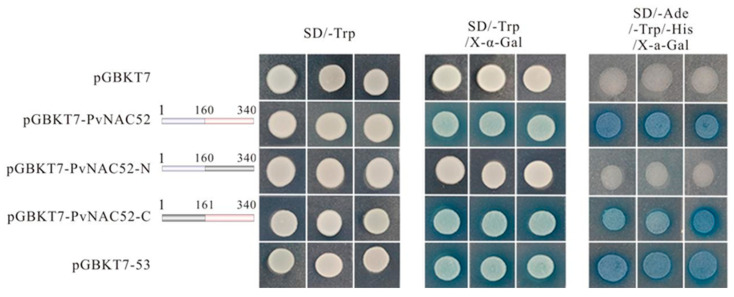
Analysis of PvNAC52 transcriptional self-activation activity. Transcriptional activity evaluation of yeast ZmNAC071: full-length (pGBKT7-PvNAC52) and partial deletion (PGBKT7-PvNAC52-N, PGBKT7-PvNAC52-C). The number indicates the amino acid position. The transformation was detected using SD/-Trp (growth control), SD/-Trp/X-α-Gal, and SD/-Ade/-Trp/-His/X-a-Gal media. Positive and negative controls were inverters carrying empty PGBKT7-53 and pGBKT7 plasmids.

**Figure 3 ijms-25-05818-f003:**
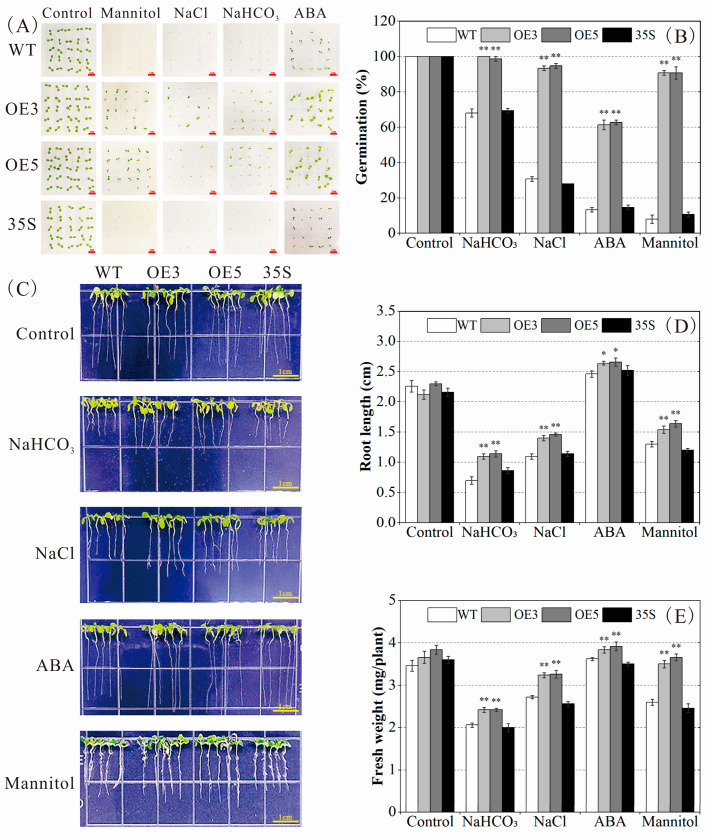
Assessment of resistance to saline-alkali, ABA, and mannitol stress under medium conditions. (**A**,**B**) Detection of seed germination. In 1/2 MS medium containing 120 mM NaCl, 7 mM NaHCO_3_, 1.5 mM ABA, and 300 mM mannitol, seeds of different lines were cultured for 8 days. Representative seedlings are shown in the photographs (*n* = 25). (**C**) Comparison of growth phenotypes of seedlings. Seedlings of different lines were cultivated under normal conditions for 7 days and transplanted into MS medium containing 120 mM NaCl, 7 mM NaHCO_3_, 1.5 μM ABA, and 300 mM mannitol for 1 week. The images were captured following 1 week of vertical cultivation. Bars = 1.0 cm. (**D**,**E**) Root length and fresh weight statistics for plants (*n* = 30). OE: *PCNAC52*-overexpressed lines; WT: wild-type lines; 35S: empty pBI121-EGFP vector control lines. Data are presented as the mean ± standard. For every treatment condition, Student’s *t*-test was used to compare the significant differences between the WT line and other lines, * *p* < 0.05, ** *p* < 0.01.

**Figure 4 ijms-25-05818-f004:**
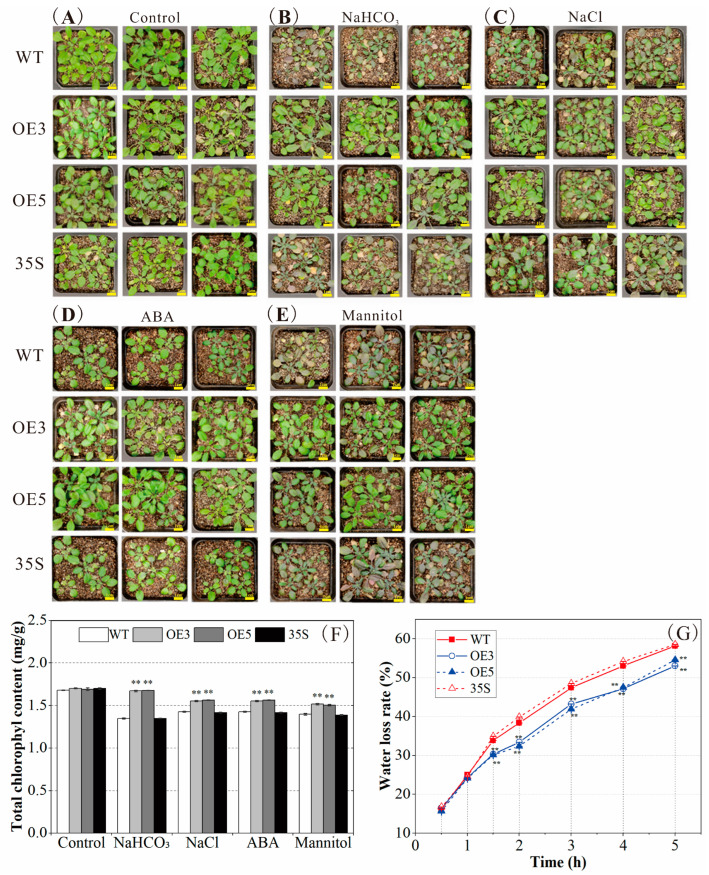
Assessment of resistance to saline-alkali, ABA, and mannitol stress under soil cultivation conditions. (**A**–**E**) Soil seedling tolerance analysis. The seedlings of each line were exposed to 180 mM NaCl, 10 mM NaHCO_3_, 50 μM ABA, and 300 mM mannitol for 16 days (*n* = 100) and subsequently photographed. Bars = 1.0 cm. (**F**) Chlorophyll content analysis. After being treated with NaCl (180 mM), NaHCO_3_ (10 mM), ABA (50 μM), and mannitol (300 mM) for 16 days, the chlorophyll contents of seedlings of different lines were determined. (**G**) Determination of water loss rates of different lines under normal culture conditions. OE: *PCNAC52*-overexpressed lines; WT: wild-type lines; 35S: empty pBI121-EGFP vector control lines. Data are presented as the mean ± standard. For every treatment condition, Student’s *t*-test was used to compare the significant differences between the WT line and other lines, ** *p* < 0.01.

**Figure 5 ijms-25-05818-f005:**
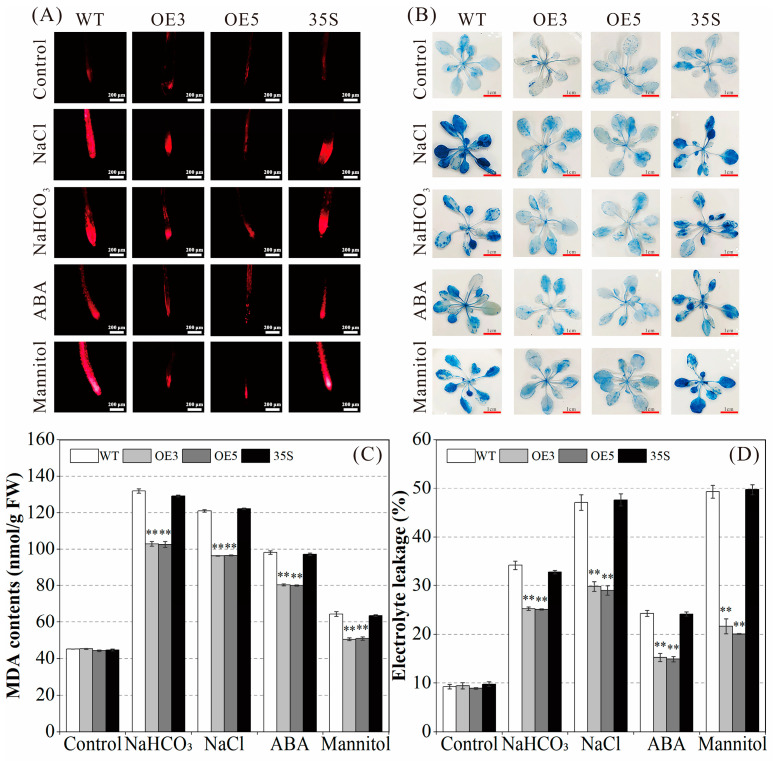
Analysis of lipid peroxidation of cell membrane. (**A**) Propidium iodide (PI) staining was used to observe cell membrane damage at plants’ root tips under different stress treatments. Bars: 200 μm. (**B**) The cell membrane damage of the leaves of each plant under different stress treatments was observed by Evans Blue staining. Bars: 1 cm. (**C**,**D**) Malondialdehyde (MDA) content and conductivity of each line under different stress treatments. OE: *PCNAC52*-overexpressed lines; WT: wild-type lines; 35S: empty pBI121-EGFP vector control lines. Data are presented as the mean ± standard. For every treatment condition, Student’s *t*-test was used to compare the significant differences between the WT line and other lines, ** *p* < 0.01.

**Figure 6 ijms-25-05818-f006:**
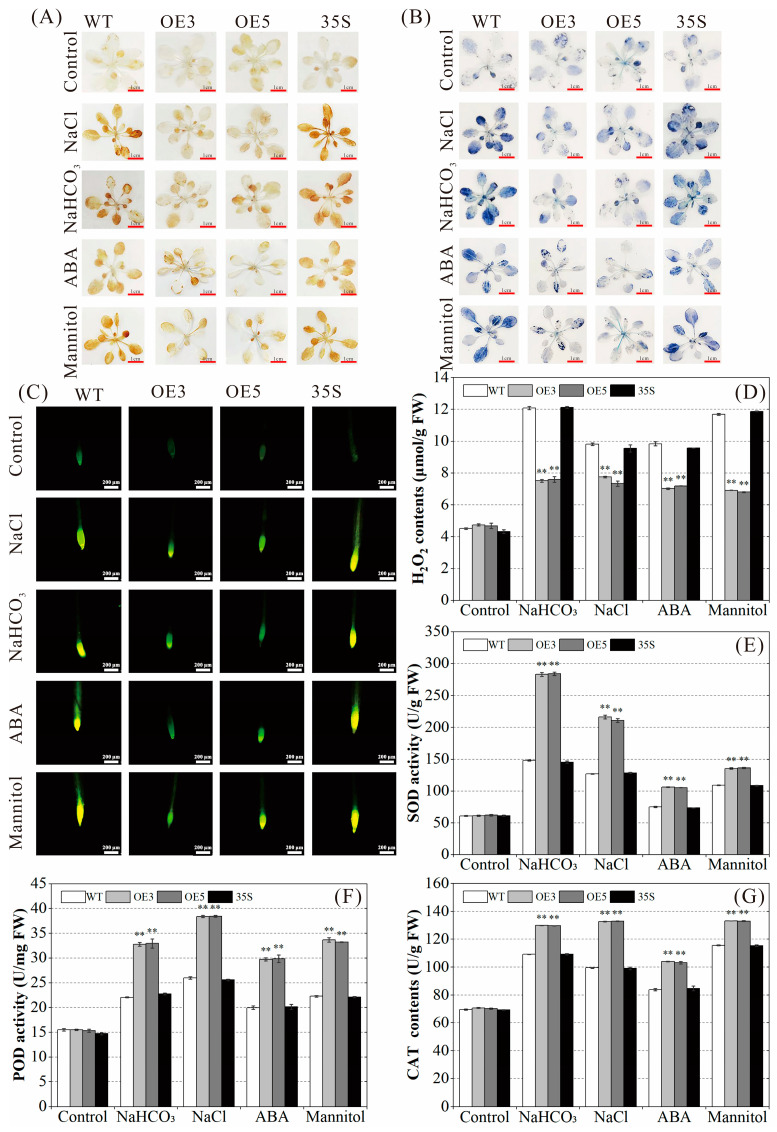
Detection of ROS accumulation and removal capacity. (**A**,**B**) ROS accumulation, analyzed using NBT and DAB staining. (**C**) H_2_DCF-DA staining was employed to examine the H_2_O_2_ content in root tips. (**D**) Detection of H_2_O_2_ content of each lines plant under different stress conditions. (**E**–**G**) CAT, POD, and SOD activities in different lines under different stress conditions. OE: *PCNAC52*-overexpressed lines; WT: wild-type lines; 35S: empty pBI121-EGFP vector control lines. Data are presented as the mean ± standard. For every treatment condition, Student’s *t*-test was used to compare the significant differences between the WT line and other lines, ** *p* < 0.01.

**Figure 7 ijms-25-05818-f007:**
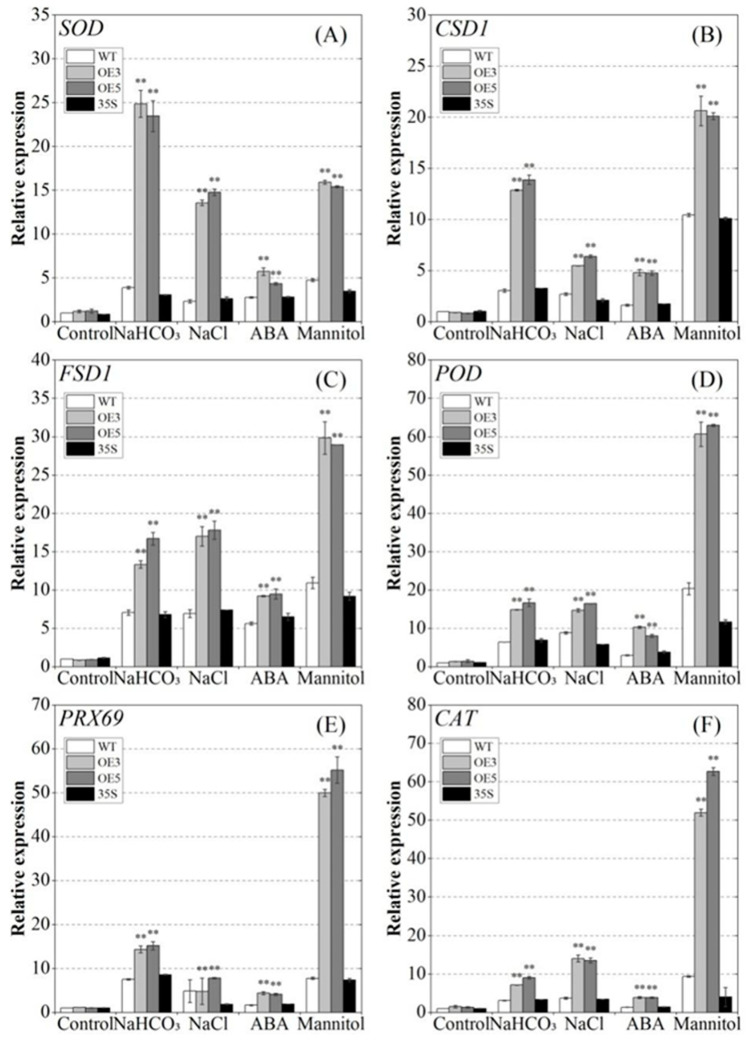
Salt-alkali, ABA, and mannitol treatment effects on SOD, POD, and CAT related gene expression. (**A**–**F**) *SOD*, *CSD1*, *FSD1*, *POD*, *PRX69*, and *CAT* genes expression. OE: *PCNAC52*-overexpressed lines; WT: wild-type lines; 35S: empty pBI121-EGFP vector control lines. The gene expression in the WT plant was set to one. Data are presented as the mean ± standard. For every treatment condition, Student’s *t*-test was used to compare the significant differences between the WT line and other lines, ** *p* < 0.01.

**Figure 8 ijms-25-05818-f008:**
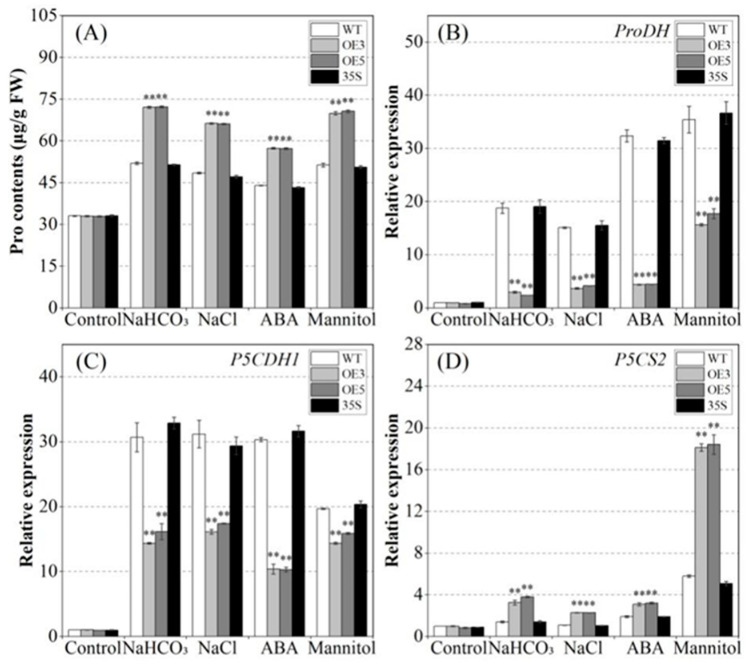
Regulation analysis of the *PvNAC52* overexpression effect on proline content. (**A**) Changes in proline content in different lines under various types of stress. (**B**–**D**) Proline synthesis and degradation genes expressed in each line under different stresses. Pro: proline; OE: *PCNAC52*-overexpressed lines; WT: wild-type lines; 35S: empty pBI121-EGFP vector control lines. The gene expression in the WT plant was set to one. Data are presented as the mean ± standard. For every treatment condition, Student’s *t*-test was used to compare the significant differences between the WT line and other lines, ** *p* < 0.01.

**Figure 9 ijms-25-05818-f009:**
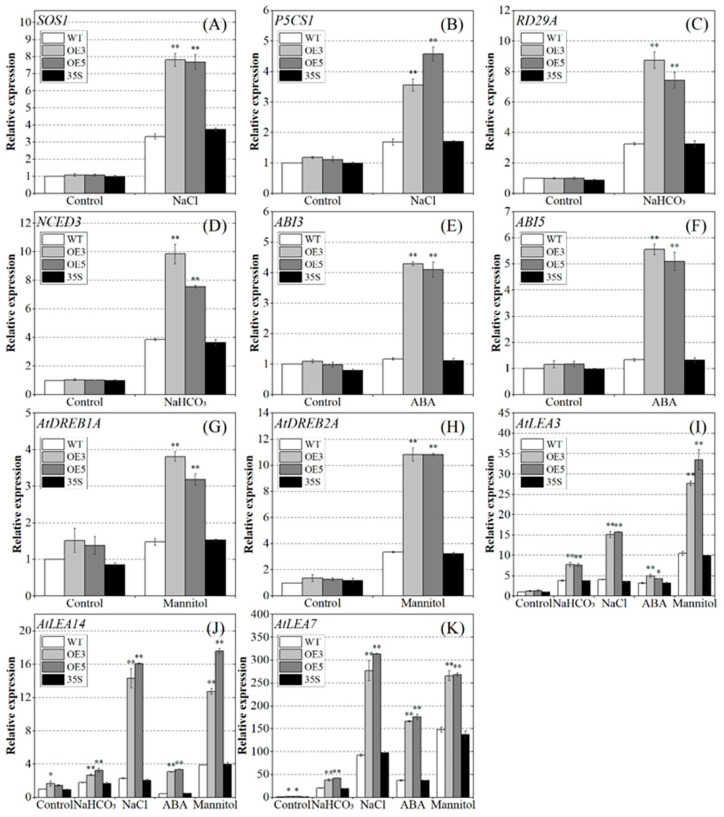
Expression of stress-associated genes in diverse lines treated with saline alkali, ABA, and mannitol. (**A**–**K**) *SOS1*, *P5CS1*, *RD29A*, *NCED3*, *ABI3*, *ABI5*, *AtDREB1A*, *AtDREB12A, AtLEA3*, *AtLEA14*, and *AtLEA7* genes expression. OE: *PCNAC52*-overexpressed lines; WT: wild-type lines; 35S: empty pBI121-EGFP vector control lines. The gene expression in the WT plant was set to one. Data are presented as the mean ± standard. For every treatment condition, Student’s *t*-test was used to compare the significant differences between the WT line and other lines, * *p* < 0.05, ** *p* < 0.01.

**Figure 10 ijms-25-05818-f010:**
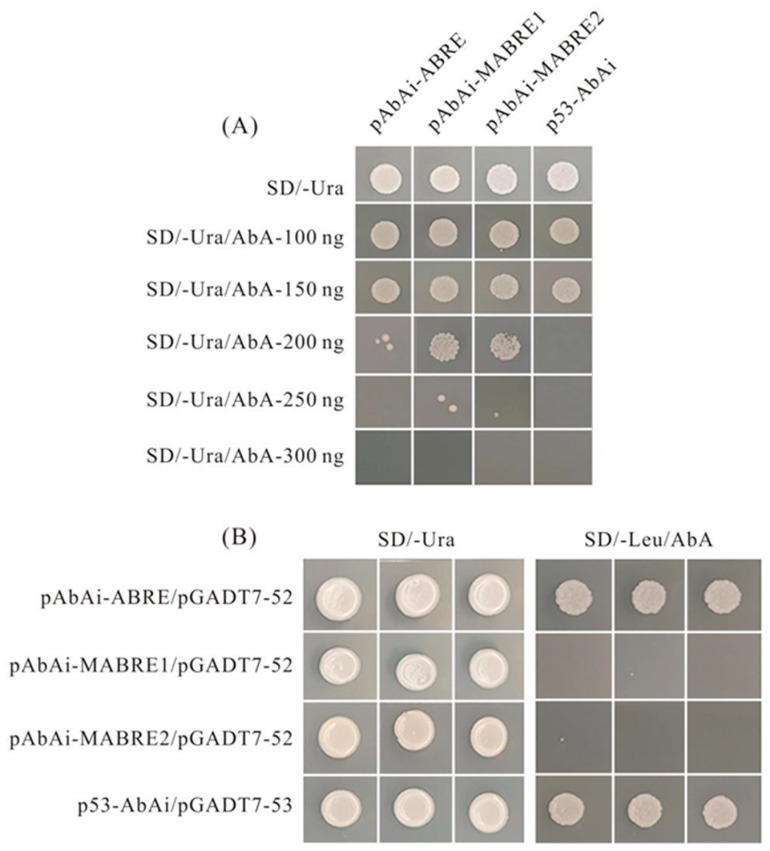
Binding specificity of *PvNAC52* and *PvNAC52* promoter ABRE elements. (**A**) Validation of Aureobasidin A (AbA) resistance of bait strains. (**B**) Validation of *PvNAC52* in combination with ABRE components.

**Figure 11 ijms-25-05818-f011:**
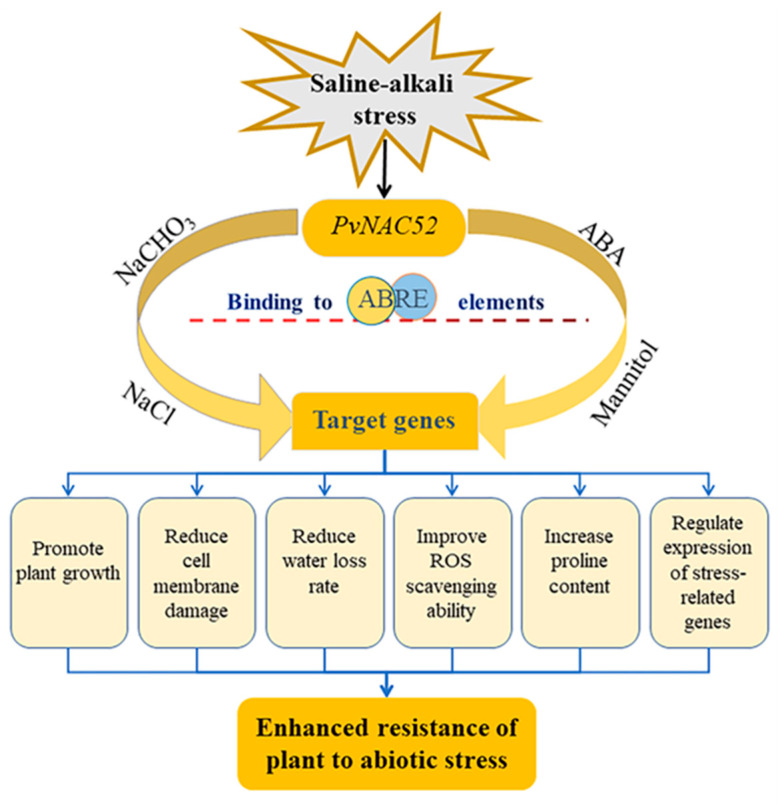
Working model diagram of *PvNAC52* in abiotic stresses.

## Data Availability

The authors confirm that the data supporting the findings of this study are available in the article. Further inquiries can be directed to the corresponding authors.

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
