# Peer review of "Common Bean (Phaseolus vulgaris L.) NAC Transcriptional Factor PvNAC52 Enhances Transgenic Arabidopsis Resistance to Salt, Alkali, Osmotic, and ABA Stress by Upregulating Stress-Responsive Genes"

_ijms, 2024, doi:10.3390/ijms25115818_

Round 1
Reviewer 1 Report
Comments and Suggestions for Authors
The paper performs a complete and deep study on a previously uncharctrized transcription factor from a major pulse as is common bean, one of the most important source of proteins in developing countries. The science is sound and the characterization has been meticulously performed. I only have some minor comments on this paper.
Statistics: in all the figures the asterisks indicate the p-value, but is not explicitly mentioned in each figure which analysis was performed (student's test) and ehich bar was used to compare (control without stress in all the bars, or the control with stress for each set of data), Please, include this information in each figure legend.
Figure 1: The nuclear localization is hardly observed. A stain with a bona fide nuclear marker such as DAPI is required.
Figure 3A: for coherence with the other figures and pannels, control should be located to the left.
Figure 3 and others: Please revise carefully the titles in the axis, as there are many misspellings (Contronl; Manntiol...).
Figure 3: is too dense and plants are hardly seen. I recommend to split the figure and made pannels F, G and H as a separate figure for the sake of clarity.
Line 381: a stop is missing before However.
Reviewer 2 Report
Comments and Suggestions for Authors
- This paper correspond for scope of journal.
- The title corresponds to the content of the paper.
- This paper represents a significant contribution to improve knowledge about functional mechanisms of transcription factors (TF NAC) for increasing adaptability of bean plants to abiotic stress conditions, a transcriptional activation potential in the C-terminal region, which regulates downstream target genes by binding to the ABER cis-acting element, and also in estimation role of the PvNAC52 protein for plant resilience to salt, alkali, osmotic, and ABA stresses.
- The main question of research is addressed to estimate effects of PvNAC52 to SOD, POD, and CAT gene expression in the OE plant than that in the WT and 35S plants under different stress conditions as well as detection of H2O2 content, proline contents, of each lines plant under different stress conditions, and activities of CAT, POD, and SOD in different lines under different stress.
- Also, effect of PvNAC52 to seed development, the germination rate, seedling root length, whole plant fresh weight, and chlorophyll content of rosette leaves of Arabidopsis plants were studied under different stress
-
- The aim of research is not pointed out. Aim of research is necessary write as last paragraph of chapter of Introduction. Shoud be write what was the aaim of study (“example: the aim was functional caracterisation of novel activator PvNAC52 screened from the common bean 'Qingyun No. 1' and how the common bean NAC TFs modulate plant responses to diverse abiotic stresses, including salt-alkali, ABA, and osmotic stresses….)”
- Key words are appropriate.
- For study used adequate methods. However the chapter 4. Materials and Methods should be positioned after chapter Introduction, and accordingly numerates as chapter 2.
-
- Results are clearly presented and discussed.
- Tables, figures, pictures are clear.
- Conclusions are written based on the results obtained in this research
- Manuscript is acceptable after minor corrections.
Suggestion:
The order of the chapters is not numbered correctly. The chapter 4. Materials and Methods should be should be numbered as chapter 2 and positioned after chapter Introduction.
It is also necessary to renumber the Results chapter as chapter 3 instead of the assigned number 2 to this version of the paper, and to number the discussion as chapter 4.
In accordance with this chapter numbering, renumber the subchapters.
On line 41 should be correct error and change word “…..unfacourable…..” with “unfavourable”
